# Rescue of Mitochondrial SIRT3 Ameliorates Ischemia-like Injury in Human Endothelial Cells

**DOI:** 10.3390/ijms23169118

**Published:** 2022-08-14

**Authors:** Xi Liu, Yi Li, Zhen Zhang, Juan Lu, Gang Pei, Shichao Huang

**Affiliations:** 1State Key Laboratory of Cell Biology, Shanghai Institute of Biochemistry and Cell Biology, Center for Excellence in Molecular Cell Science, Chinese Academy of Sciences, Shanghai 200031, China; 2Shanghai Key Laboratory of Signaling and Disease Research, Laboratory of Receptor-Based Biomedicine, The Collaborative Innovation Center for Brain Science, School of Life Sciences and Technology, Tongji University, Shanghai 200070, China; 3Institute for Stem Cell and Regeneration, Chinese Academy of Sciences, Beijing 100045, China

**Keywords:** cell death, endothelial cells, Sirtuin 3 (SIRT3), OGD/R, ischemic stroke

## Abstract

Structural and functional alterations of vasculature caused by age-related factors is critically involved in the pathogenesis of ischemic stroke. The longevity genes sirtuins (SIRTs) are extensively investigated in aging-associated pathologies, but their distinct roles in ischemic stroke still remain to be clarified. To address this question, we applied oxygen and glucose deprived/reperfusion (OGD/R) to induce ischemic injury in human endothelial cells (ECs), which are the main component of vasculature in the brain. The results showed that OGD/R led to various damages to ECs, including compromised cell viability, increased LDH release, overproduced ROS, enhanced apoptosis and caspase activity. Meanwhile, the expression of mitochondrial SIRT3 was robustly decreased in ECs after OGD/R treatment. Consistently, rescue of SIRT3 by ectopic expression, but not nuclear SIRT1, in ECs reversed the OGD/R-induced cell damage. Interestingly, some front-line drugs for ischemic stroke, including clopidogrel, aspirin and dl-3-n-butylphthalide (NBP), also rescued SIRT3 and reduced OGD/R-induced endothelial injury, suggesting that the recovery of SIRT3 expression was critical for the protection of these drugs. Moreover, our results demonstrated that 10-hydroxy-NBP (OHNBP), a major metabolite of NBP, showed better blood-brain barrier crossing capability than NBP, but still retained the effects on SIRT3 by NBP. Together, our results suggested that SIRT3 may serve as a potential novel target for treatment of ischemic stroke.

## 1. Introduction

Ischemic stroke is a leading cause of death and long-term disability, and is usually caused by the occlusion of cerebrovascular vessels, which leads to an interrupted or reduced supply of blood to the brain and induces the necrosis of brain tissues. Currently, thrombolysis using intravenous recombinant tissue plasminogen activator (rtPA) is the only therapy approved by the FDA for ischemic stroke, while it can only be delivered to a small fraction of patients due to the narrow treatment window [1]. Other treatments including antiplatelet (such as clopidogrel, aspirin), neuroprotection (such as NBP) and antioxidant agents (such as edaravone) are also implemented in clinical practice, which provide limited benefits to patients [2,3,4,5,6,7]. A better understanding of stroke pathology is important to identify innovative targets to broaden our therapeutic options in this scenario.

Aging is the most robust risk factor for ischemic stroke, which doubles every 10 years after age 55 years. Approximately three-quarters of all strokes occur in persons aged ≥65 years [8]. With aging, cerebral vasculature undergo structural and functional alterations, which further increases susceptibility to ischemia [9,10]. Endothelial cells are the main component of vasculature and the blood-brain barrier, which critically control vascular permeability and infiltration of inflammatory mediators [9,11]. Accordingly, endothelial injury amplifies irreversible neuronal death and worsened parenchymal damage [9,10,12]. With the pivotal role of endothelial cells in the development of ischemic stroke, it is necessary to investigate it as a promising interventional target to blunt ischemia-induced injury. 

Several proteins regulating aging are shown to determine cerebral damage in ischemic injury. Specifically, sirtuins are NAD^+^-dependent deacetylases with a broad range of biological processes, such as aging, cell metabolism, inflammation and apoptosis, which are key to the stroke pathophysiology [13]. SIRT3 is involved in mitochondrial function and biosynthetic pathways such as fatty acid metabolism, oxidative stress and apoptosis [13,14]. Previous study has demonstrated that SIRT3 downregulation increased the susceptibility of cardiac-derived cells to ischemia-reperfusion injury and contributed to worse injury in the aged heart [15,16]. SIRT3 knockout led to coronary microvascular dysfunction and impaired cardiac recovery post myocardial ischemia [15,16,17,18,19]. Thus far, a possible role of SIRT3 has been investigated only for neuronal and astrocytic SIRT3 in ischemic stroke [20,21], but has not been explored in endothelial cells. 

In view of the important role of endothelial cells in microvascular circulations, we hypothesized that the endothelial expression of the longevity gene SIRT3 may be protective in ischemia-like models. Here, we verified that SIRT3, but not SIRT1, decreased in ECs damaged by OGD/R. Furthermore, we investigated whether rescue of SIRT3 expression in ECs could reverse the OGD/R-induced damage. Furthermore, we tested whether the pharmacological phenotype of clinical drugs, such as clopidogrel (Clo), aspirin (Asp), NBP and edaravone (ED), was linked to the target SIRT3. Finally, we optimized NBP to enhance its ability to cross the blood-brain barrier, and retain the protective effects on SIRT3 by NBP.

## 2. Results

### 2.1. OGD/R Induces Injury in ECs

OGD/R could induce EC injury including decreased cell viability, increased LDH release, apoptosis and promoted ROS production. Here, we treated ECs with OGD for 2 h, 4 h or 6 h and measured cell viability after reperfusion at 24 h using CCK-8 or CTG methods. We observed that OGD/R induced decreased cell viability, increased LDH release both in HUVEC cells (Figure 1A–C) and HBMEC cells (Figure 1G–I). Data showed that the cell viability of ECs exposed to OGD/R decreased dependently with duration, and OGD for 4 h in HUVECs or OGD for 6 h in HBMECs significantly reduced cell viability by about 50% compared to the control (Figure 1A,B,G,H). Thus, 4 h OGD for HUVECs and 6 h OGD for HBMECs were applied for subsequent experiments. Under these conditions, we also observed ODG/R-induced apoptosis in HUVECs (Figure 1D,E) and in HBMECs (Figure 1J,K) to different extents. In addition, OGD/R induced ROS production by about 250% of the control in HUVECs (Figure 1F). These results indicated that OGD/R could mimic ischemia in vitro with similar pathology.

### 2.2. SIRT3 Expression Is Reduced in ECs after Exposure to OGD/R

The increasing evidence of sirtuins in the field of age-related diseases, such as ischemic stroke, indicates that they may provide novel targets for treating ischemic stroke [22]. First, we analyzed the published sequence data of profiling brain cortex from transient middle cerebral artery occlusion (tMCAO) rat stroke model (data from Series GSE38037) [23], and found that SIRT1, SIRT2 and SIRT7 had no significant change, while SIRT3 significantly decreased in the brain cortex of the SD rats following tMCAO compared to the sham-operated control (Figure 2A). Then, we detected the mRNA levels of SIRT3, SIRT1 and eNOS in HUVECs after exposure to OGD/R. As expected, the mRNA levels of SIRT3, but not SIRT1, decreased to about 50% of the control (Figure 2B,C). We also observed that eNOS decreased in OGD/R-treated HUVECs (Figure 2D), which indicated endothelial dysfunction. In addition, we determined the protein levels of the above genes, and observed similar changes of these proteins in HUVECs (Figure 2E–I) and HBMECs (Figure 2J–M). All these results suggested that SIRT3 robustly decreased in ECs induced by OGD/R. 

### 2.3. Rescue of SIRT3 Expression Reverses ODG/R-Induced Cell Damage in ECs

According to the above results, we speculated that high SIRT3 or SIRT1 expression could rescue OGD/R-induced injury. Although SIRT1 had no significant effect in OGD/R-treated ECs, we still want to know if SIRT1 overexpression in ECs could delay aging and decrease endothelial susceptibility to ischemia-like injury. Thus, we overexpressed SIRT3 or SIRT1 in ECs. Western blotting analysis was performed from SIRT3^OE^ cells and revealed approximately 2-fold increases of total SIRT3 protein, compared with the vector group (Figure 3A). Then, the CTG assay was applied to evaluate the effect of SIRT3 on the cell viability in HUVECs after exposed to OGD/R. The data showed that SIRT3 overexpression in HUVECs rescued the cell viability of the cells treated with OGD/R, while SIRT1 overexpression did not exhibit this effect (Figure 3B). We also observed the same phenomenon in HBMECs (Figure 3L,M). Besides, we confirmed that SIRT3 overexpression inhibited the apoptosis of HUVECs after exposure to OGD/R (Figure 3C–G,J–K). Moreover, we found that SIRT3 overexpression potently rescued OGD/R-induced ROS overproduction in HUVECs (Figure 3H,I). Taken together, ectopic expression of SIRT3, but not SIRT1, was shown to reverse ODG/R-induced cell damage in ECs.

### 2.4. Clinical Drugs Rescue SIRT3 to Reduce OGD/R-Induced Injury

To further identify SIRT3 as a target in this setting, a subsequent step whether the pharmacological phenotype of clinical drugs, such as clopidogrel (Clo), aspirin (Asp), dl-3-n-butylphthalide (NBP) and edaravone (ED), is actually linked to the target SIRT3 is required. First, we performed Western blotting analysis to determine how these clinical drugs affected SIRT3 in ECs after exposure to OGD/R. As shown in Figure 4A, NBP upregulated SIRT3 to the maximum extent at 3 μM and slightly down at 10 μM in OGD/R damaged HUVECs, which was consistent with the phenotype of cell viability and LDH release (Figure 4B,C). Similarly, Clo and Asp also upregulated SIRT3 dose-dependently in positive correlation with cell viability (Figure 4A–C). However, this link between the phenotype and upregulated SIRT3 was not seen in ED, which could not upregulate SIRT3 but had positive effects on the cell viability in OGD/R-damaged HUVECs (Appendix A). In addition, this correlation was observed not only in HUVECs but also in HBMECs (Figure 4D,E). In conclusion, these results indicated that the extent of SIRT3 upregulated was highly correlated with the pharmacological properties of these clinical drugs.

Furthermore, we asked whether these findings could be applied in improving the clinical drugs for ischemic stroke, such as NBP. Based on our findings, we hypothesized that NBP derivatives, which showed better blood-brain barrier crossing capability and retained the enhancing effects on SIRT3, might be considered as promising drug candidates for ischemic stroke. Furthermore, 10-hydroxy-NBP (OHNBP), a major circulating metabolite of NBP, was reported to have a much greater ability to cross the blood-brain barrier than NBP [24,25,26]. Thus, we detected the distribution of OHNBP in plasma following oral administration to rats with 20 mg/kg NBP or OHNBP and showed no statistically significant difference between the two group during 8 h, but in the brain, it was increased about 2-fold at 0.5 h and about 3-fold at 8 h compared to that of the NBP group (Figure 4G,H). Furthermore, we tested whether OHNBP still retained the effects on SIRT3 as NBP. As shown in Figure 4I–J, OHNBP could still upregulate SIRT3 expression dose-dependently in OGD/R-treated HUVECs, similarly to that exhibited by NBP. Furthermore, OHNBP alleviated OGD/R-induced injury including enhanced cell viability and decreased LDH release in HUVECs, similar to NBP with no significant change (Figure 4K,L and Appendix A). This correlation between SIRT3 upregulation and decreased cell injury regulated by OHNBP was also observed in HBMECs (Appendix A). These results suggested that OHNBP may serve as a better alternative to replace NBP due to its similar protective effects against endothelial injury but better blood-brain barrier crossing capability. 

NBP has multiple pharmacologic effects in ischemic stroke, not only of endothelial protection but also of neuroprotection, anti-oxidation and anti-platelet aggregation [2,7]. In order to verify our hypothesis, we further compared the protective role of OHNBP and NBP in neuroprotection, anti-oxidation and anti-platelet aggregation. In neuroprotection, OHNBP increased cell viability of SK-N-SH cells induced by OGD/R, with no significant change compared to NBP (Appendix A). In anti-oxidation, OHNBP had the same effect in protecting SK-N-SH cells damaged by H_2_O_2_, compared to NBP (Appendix A). In anti-platelet aggregation, OHNBP inhibited platelet aggregation induced by AA similar to NBP, but showed stronger anti-platelet aggregation induced by collagen compared to NBP (Appendix A). NBP, OHNBP and Asp could not inhibit ADP-induced platelet aggregation (Appendix A). All these results indicated that OHNBP had equal or even better pharmacology effect in vitro and may serve as an alternative interventional option in place of NBP for patients suffering from ischemic stroke, which need to be further demonstrated in vivo.

## 3. Discussion

Aging-induced circulatory alterations promoted the occurrence of ischemia. In order to explore a promising interventional target, efforts have been made to delay aging to decrease ischemia-induced injury. SIRT1 and SIRT3 are extensively investigated as longevity proteins, which delay the process of aging. In our work, we compared the role of these two proteins in ischemia-associated endothelial injury and unexpectedly found that mitochondrial SIRT3, but not nuclear SIRT1, could prevent OGD/R-induced endothelial injury. As already known, mitochondrion is directly damaged by the initial burst of ROS production upon ischemia-reperfusion [27]. Thus, preventing mitochondria damage is a promising therapeutic strategy against ischemia-reperfusion injury. Given the critical role of mitochondria in ischemia, SIRT3 is able to exert ischemia-protecting effects through its direct modulation of mitochondrial function, such as ROS detoxification and tricarboxylic acid cycle [28]. Taken together, these findings suggested that mitochondrial SIRT3 may serve as a better potential target than nuclear SIRT1 to treat ischemic injury. Furthermore, we will investigate to what extent SIRT3 can protect mitochondrion and how SIRT3 exerts the modulation of mitochondrial function in ischemia-like injury.

In our work, we also used clinical drugs to verify the specific therapeutic relevance with SIRT3. We found that the induction of longevity-promoting SIRT3 was important for the protective effects against ischemia injury of clinical drugs including Clo, Asp and NBP. We also showed that ED ameliorated OGD/R-induced injury without upregulating SIRT3, suggesting that ED was not likely to exert it effects through SIRT3. These findings give us a hint that we could combine SIRT3-enhancing drugs with other symptomatic treatments, such as NBP and ED, to increase their therapeutic efficacy in age-dependent disease, such as ischemia. Therefore, our results indicate that SIRT3 may serve as a potential therapeutic target for developing novel ischemic treatments.

## 4. Materials and Methods

### 4.1. Drugs and Reagents

Aspirin (Asp), clopidogrel (Clo) and edaravone (ED) were purchased from MedChemExpress (Shanghai, China). NBP and OHNBP were synthesized by WuXi AppTec (Tianjing, China). The other reagents and materials are mentioned in each method.

### 4.2. Animals

Healthy male Sprague Dawley rats weighing 180–240 g were purchased from Beijing Vital River Experimental Animal Technology Co., Ltd. (Beijing, China). The rats were housed in a specific-pathogen-free laboratory, kept under standard environmental conditions (23 ± 1 °C, humidity 45 ± 5%, 12-h light/dark cycle) and had free access to standard food.

### 4.3. Cell Culture 

HUVEC and HBMEC cells were purchased from ATCC and BeNa Culture Collection, respectively. The two endothelial cell lines were cultured in ECM supplemented with 5% FBS, 1% ECGS and 1% penicillin/streptomycin (ScienCell Research Laboratories, San Diego, CA, USA) in a humidified incubator with 5% CO_2_/95% air (*v*/*v*) at 37 °C. SK-N-SH cells were purchased from ATCC. SK-N-SH cells were cultured in MEM supplemented with 10% FBS and 1% penicillin/streptomycin in a humidified incubator with 5% CO_2_/95% air (*v*/*v*) at 37 °C.

### 4.4. OGD/R-Induced Cell Injury Model

Approximately 4 × 10^3^ HUVEC, 4 × 10^3^ HBMEC or 4 × 10^3^ SK-N-SH cells were cultured in 96-well plates under normal conditions for 24 h. After pretreated with or without NBP, OHNBP, Clo, Asp or ED (1, 3, 10 μM) for 2 h, the culture was replaced by glucose-free DMEM (Gibco, New York, NY, USA) and placed under hypoxia conditions (AnaeroPack, #D-07, MGC, Chiyoda-ku, Tokyo, Japan) at 37 °C for applied time periods. The medium was then replaced by normal medium without FBS and reoxygenated under normoxic conditions (95% air, 5% CO_2_) for another 24 h to induce OGD/R injury. Cells incubated in normal medium without FBS under a normoxic atmosphere were used as the control.

### 4.5. H_2_O_2_-Induced Cell Injury Model

Furthermore, 1 mM H_2_O_2_ was added to the SK-N-SH cells for 1 h or 2 h with or without pretreatment with NBP or OHNBP for 2 h. The SK-N-SH cells were then cultured with or without NBP or OHNBP for another 24 h to detect cell viability.

### 4.6. Cell Viability

Cell viability was detected using Cell Titer-Glo Luminescent Assay (CTG) (Vazyme Biotech, Nanjing, China) or the Cell Counting Kit-8 (CCK-8) (Beyotime Biotechnology, Shanghai, China), and all experiments were performed according to the manufacturer’s instructions. Luminescence or absorbance were measured by BioTek SynergyNEO (BioTek, Winusky, VT, USA).

### 4.7. Lactate Dehydrogenase (LDH) Release Assay

The release of cytoplasmic LDH indicates the loss of cell membrane integrity, which represents the death of the cell. We used the commercial LDH-Cytotoxicity Colorimetric Assay Kit purchased from Biovison (#K311, Cambridge, UK), according to the manufacturer’s instructions. In brief, 50 μL supernatant from each cell culture well was collected for a coupled enzymatic reaction in which LDH catalyzed the conversion of lactate to pyruvate via the reduction of NAD+ to NADH for 30 min at room temperature. It was then measured spectrophotometrically at 490 nm with a BioTek SynergyNEO (BioTek, Winooski, VT, USA). 

### 4.8. Cell Apoptosis Assay with Flow Cytometry

Cell apoptosis was detected using an Annexin V-FITC/propidium iodide (PI) apoptosis assay kit (#556547, BD Pharmingen, Lake Franklin, NJ, USA). HUVECs or HBMECs were cultured in 12-well plates and harvested after OGD/R treatment. The cells were then stained with Annexin V-FITC and PI according to the manufacturer’s instructions and analyzed using flow cytometry. The total apoptosis rate (%) was calculated as (Annexin V^+^ PI^−^ cells + Annexin V^+^ PI^+^ cells)/total number of cells × 100. The early apoptosis rate (%) was calculated as Annexin V^+^ PI^−^ cells /total number of cells × 100. The late apoptosis rate (%) was calculated as Annexin V^+^ PI^+^ cells /total number of cells × 100.

### 4.9. ROS Detection

To detect intracellular ROS levels, 2′,7′-Dichlorodihydrofluorescein diacetate (DCFH-DA) (Beyotime, Shanghai, China) was used as a probe. Following OGD/R treatment, HUVEC cells were incubated with 10 μM DCFH-DA in culture medium without FBS at 37 °C for 30 min. The cells were washed with PBS twice and then captured by a fluorescence microscope (Leica, Wetzlar, Germany). Alternatively, the fluorescence intensity of the cells in the 96-well black plate after exposure to OGD/R was measured using a BioTek SynergyNEO (BioTek, USA) at an excitation wavelength of 488 nm and an emission wavelength of 525 nm.

### 4.10. Plasmids and Lentiviral Constructs and Infection

For SIRT3 or SIRT1 overexpression experiments, SIRT3 pcDNA3.1 or SIRT1 pcDNA3.1 were cloned into a Fugw vector. Lentivirus packing and infection were conducted as follows. HEK293T cells were seeded at a density of 7.5 × 10^6^ cells in 100-mm dishes. On the following day, cells were transfected with 10 μg Fugw-SIRT3 constructs, 7.5 μg of pSPAX2 and 5 μg of pMD2G via PEI. The virus-containing supernatant was collected at 48 h and 72 h, respectively, and it was then centrifuged at 1000× *g* for 5 min, before being passed through 0.45 μm filters. The lentivirus was further concentrated by ultracentrifugation at 27,000× *g* for 2 h. The pellets were then resuspended in 200 μL PBS, aliquoted and stored at −80 °C. For SIRT3 or SIRT1 overexpression experiments, HUVECs or HBMECs were seeded in 6-well plates before concentrated lentivirus infection (minimum multiplicity of infection) in the presence of polybrene (6 μg/mL, Sigma, St. Louis, MO, USA). After 12 h, the medium was refreshed. The expression of SIRT3 or SIRT1 was determined by Western blotting at 72 h. 

### 4.11. Reverse Transcription and Quantitative Real-Time PCR

After treatment with OGD/R, cells were extracted by 500 μL TRIzol Reagent (#T9424, Sigma, St. Louis, MO, USA) to obtain total RNA according to the manufacturer’s instructions. Reverse transcription was conducted using PrimeScript RT master mix (#RR036B, TaKaRa, Shiga Prefecture, Japan) under the following conditions: 37 °C, 15 min and 85 °C, 15 s. Then, the 25 μL reaction system containing 4 μL prediluted cDNA, 0.25 μM of each primer and 2 × HotStart SYBR Green qPCR master mix (#MB000-3013, ExCell Bio, Suzhou, China), was analyzed using a Stratagene Mx3000P (Agilent Technologies, Santa Clara, CA, USA). The reaction parameters were as follows: 95 °C for 10 min; 95 °C for 30 s, 40 cycles; 60 °C for 30 s; and 72 °C for 30 s. An additional cycle was performed for evaluation of the primer’s dissociation curve: 95 °C for 1 min, 60 °C for 30 s and 95 °C for 30 s. GAPDH was used as an internal control. Primers used were as follows:

SIRT1, forward 5′-AGGCCACGGATAGGTCCATA-3′, 

reverse 5′-GTGGAGGTATTGTTTCCGGC-3′; 

SIRT3, forward 5′-TGGGCTTGAGAGAGTGTCGG-3′, 

reverse 5′-GAACCCTGTCTGCCATCACG-3′; 

eNOS, forward 5′-AGCACATTTGGGAATGGGGAT-3′, 

reverse 5′-AGCGGATCTTATAACTCTTGTGC-3′; 

GAPDH, forward 5′-AGATCCCTCCAAAATCAAGTGG-3′, 

reverse 5′-GGCAGAGATGATGACCCTTTT-3′.

### 4.12. Western Blotting

After treatment with OGD/R, total cell lysates were separated by 10 or 12.5% SDS-PAGE gels, and then transferred onto nitrocellulose membranes. Each membrane was blocked with 5% nonfat milk in TBST for 1 h at room temperature. Membranes were subsequently incubated with the designated primary antibodies: SIRT3 (1:1000, #5490S, Cell Signaling Technology, Danvers, MA, USA), SIRT1 (1:1000, #8469S, Cell Signaling Technology), eNOS (1:1000, #32027S, Cell Signaling Technology), procaspase-3 (1:1000, #14220T, Cell Signaling Technology), cleaved caspase-3 (1:1000, #9661T, Cell Signaling Technology) and actin (1:1000, #A2066, Sigma, St. Louis, MO, USA), at 4 °C overnight followed by horseradish peroxidase (HRP)-conjugated secondary antibody. Membranes were then incubated with an ECL substrate and visualized by mini chemiluminescent imaging and analysis system.

### 4.13. Liquid Chromatography/Mass Spectrometry (LC/MS) 

SD rats were orally administered with 20 mg/kg NBP or OHNBP dissolved in soybean oil at 2 mg/mL. Rats were sacrificed at 0.5 h, 2 h, 4 h and 8 h after administration (*n* = 3 for each timepoint), and then the brain and plasma were obtained and stored at −20 °C for LC/MS test.

### 4.14. Platelet Aggregation Assay

Venous blood anticoagulated with citric acid was obtained from our laboratory members who were healthy and had not taken any medication affecting platelet function in the previous two weeks. Platelet-rich-plasma (PRP) was prepared by centrifugation of blood at 1000 rpm × 10 min at room temperature and transferred to polypropylene tubes. To induce platelet aggregation, platelets were activated in the absence or presence of agonists. Particularly, aliquots (85 μL) of PRP were preincubated with 100 μM NBP, OHNBP or Asp for 10 min. For platelet labeling, 1 μL of anti-human CD61-FITC was added for 15 min at room temperature in the dark. The agonist was then added for 5 min at 37 °C under low shear stress conditions. The following platelet agonists were used: 1 mM arachidonic acid (AA, #10931, Sigma, St. Louis, MO, USA), 10 μg/mL collagen (#AG005K, Shanghai Guandong Biology, Shanghai, China) and 10 μM ADP (#A5285, Sigma, St. Louis, MO, USA). After incubation, samples were fixed with 400 μL of 4% PFA. Finally, platelet aggregates were analyzed by FCM as CD61-FITC positive events in the region.

### 4.15. Statistical Analysis

The data are presented as means ± SD, and were analyzed statistically with student *t* test or one-way ANOVA and Tukey’s multiple comparisons test using GraphPad Prism 9.0 (San Diego, CA, USA). A value of *p* < 0.05 was defined as statistically significant.

## Figures and Tables

**Figure 1 ijms-23-09118-f001:**
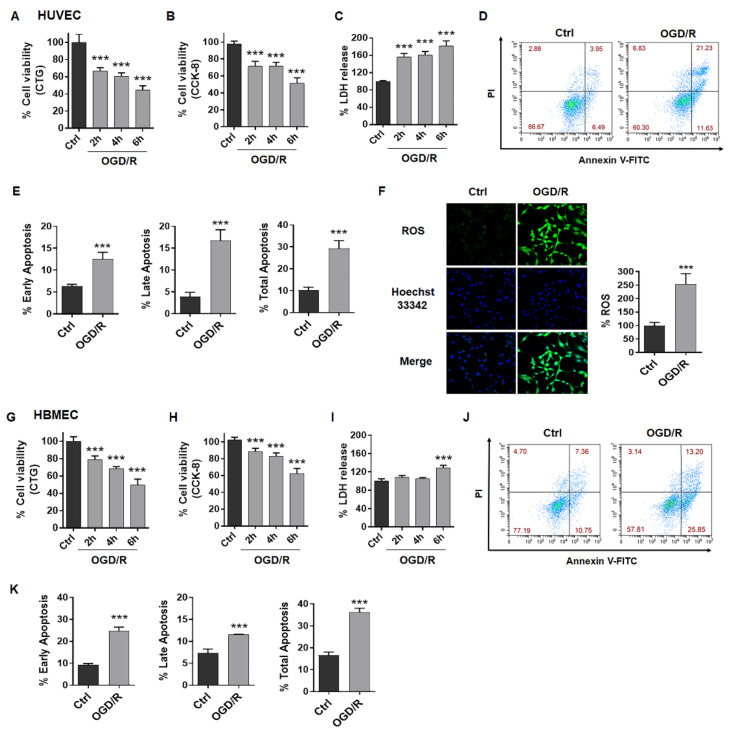
OGD/R induces injury in ECs. (**A**) Cell viability of HUVECs was tested after OGD for 2 h, 4 h or 6 h and reperfusion for 24 h using CTG assay. (**B**) Cell viability of HUVECs was tested after OGD for 2 h, 4 h or 6 h and reperfusion for 24 h using CCK-8 assay. (**C**) LDH release of HUVECs was tested after OGD for 2 h, 4 h or 6 h and reperfusion for 24 h. (**D**,**E**) FCM analysis showed apoptosis in HUVECs after OGD4h/R24h. (**F**) Images and graph showed ROS produced in OGD4h/R24h-treated HUVECs. (**G**) Cell viability of HBMECs was tested after OGD for 2 h, 4 h or 6 h and reperfusion for 24 h using CTG assay. (**H**) Cell viability of HBMECs was tested after OGD for 2 h, 4 h or 6 h and reperfusion for 24 h using CCK-8 assay. (**I**) LDH release of HBMECs was tested after OGD for 2 h, 4 h or 6 h and reperfusion for 24 h. (**J**,**K**) FCM analysis showed apoptosis in HBMECs after OGD6h/R24h. OGD/R, oxygen and glucose deprived/reperfusion. LDH, lactate dehydrogenase. HUVECs, human umbilical vein endothelial cells. HBMECs, human brain microvascular endothelial cells. The data are presented as the means ± SD, and were assessed using *t* test or one-way ANOVA. *** *p* < 0.001 vs. the Ctrl group. *n* = 3–4 for all.

**Figure 2 ijms-23-09118-f002:**
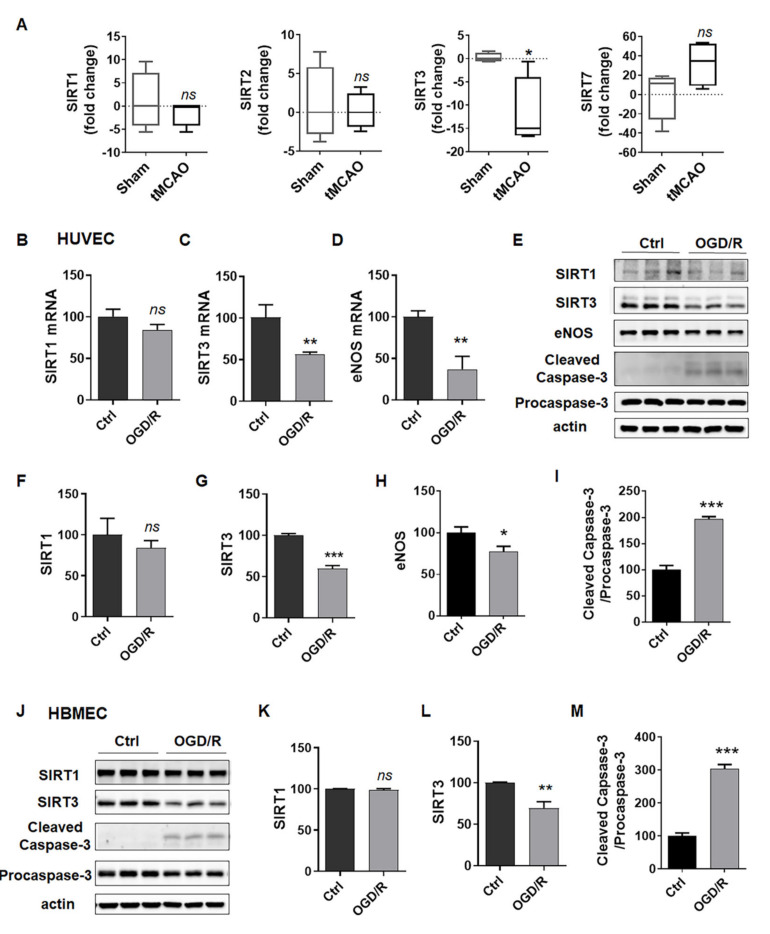
SIRT3 expression is reduced in ECs after exposure to OGD/R. (**A**) SIRT1, SIRT2, SIRT3 and SIRT7 mRNA levels in the brain cortex of the SD rats following tMCAO or the sham-operated controls from published GSE38037. (**B**–**D**) mRNA levels of SIRT1, SIRT3 and eNOS in HUVECs treated with or without OGD4h/R24h. (**E**–**I**) Protein levels of SIRT1, SIRT3, eNOS, cleaved caspase-3 and procaspase-3 in HUVECs treated with or without OGD4h/R24h. (**J**–**M**) Protein levels of SIRT1, SIRT3, cleaved caspase-3 and procaspase-3 in HBMECs treated with or without OGD6h/R24h. eNOS, endothelial nitric oxide synthase. The data are presented as the means ± SD, and were assessed using *t* test. * *p* < 0.05 vs. the Ctrl group or sham group, ** *p* < 0.01 vs. the Ctrl group, *** *p* < 0.001 vs. the Ctrl group. ns, no significance. *n* = 3 for all.

**Figure 3 ijms-23-09118-f003:**
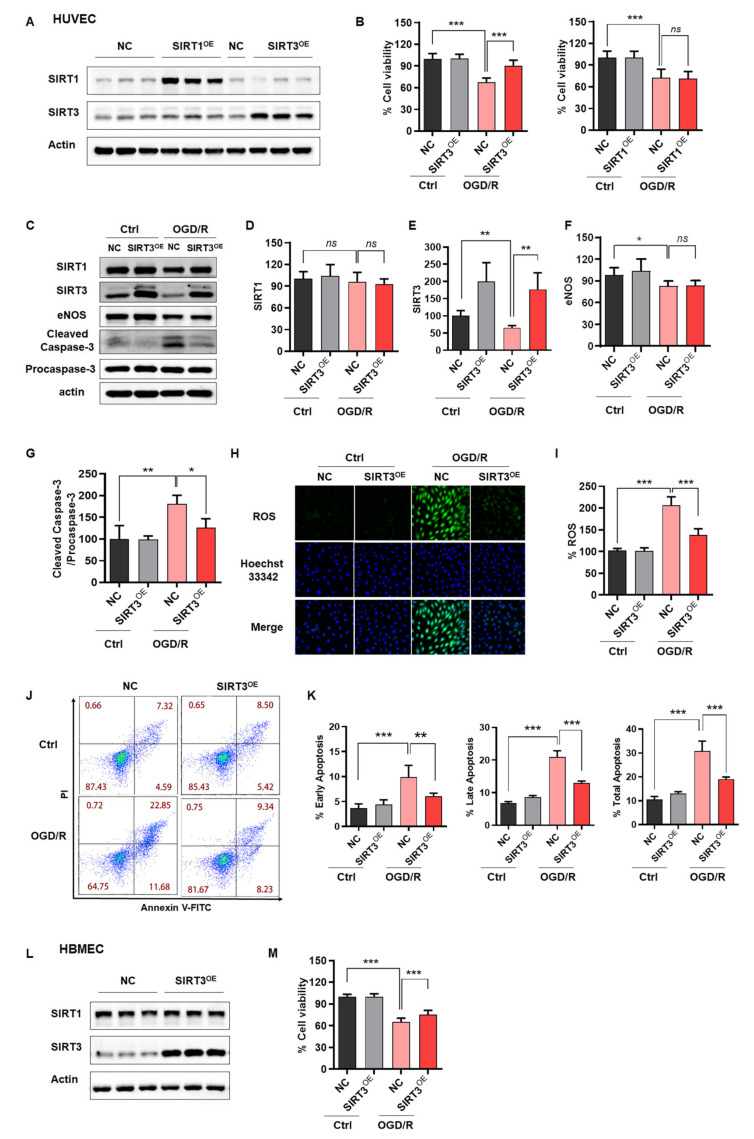
Rescue of SIRT3 expression reverses OGD/R-induced cell damage in ECs. (**A**) Western blotting verified SIRT3 and SIRT1 expression in HUVECs infected with lentivirus of SIRT3, SIRT1 or vector. (**B**) CTG assay assessed the cell viability of NC, SIRT3^OE^ or SIRT1^OE^ HUVECs under normal or OGD/R-treated conditions. (**C**–**G**) Representative immunoblot and statistical analysis of SIRT1, SIRT3, eNOS, cleaved caspase-3 and procaspase-3 from whole-cell lysates of NC or SIRT3^OE^ HUVECs under normal or OGD/R-treated conditions. (**H**,**I**) ROS levels in NC or SIRT3^OE^ HUVECs under normal or OGD/R-treated conditions. (**J**,**K**) FCM analysis showed endothelial apoptosis in NC or SIRT3^OE^ HUVECs under normal or OGD/R-treated conditions. (**L**) Western blotting verified SIRT3 expression in HBMECs infected with lentivirus of SIRT3 or vector. (**M**) CTG assay assessed the cell viability of NC, SIRT3^OE^ HBMECs under normal or OGD/R-treated conditions. NC, HUVECs or HBMECs infected with lentivirus of vector. SIRT3^OE^, HUVECs or HBMECs infected with lentivirus of SIRT3. SIRT1^OE^, HUVECs infected with lentivirus of SIRT1. The data are presented as the means ± SD, and were assessed using one-way ANOVA. * *p* < 0.05, ** *p* < 0.01, *** *p* < 0.001. ns, no significance. *n* = 3–4 for all.

**Figure 4 ijms-23-09118-f004:**
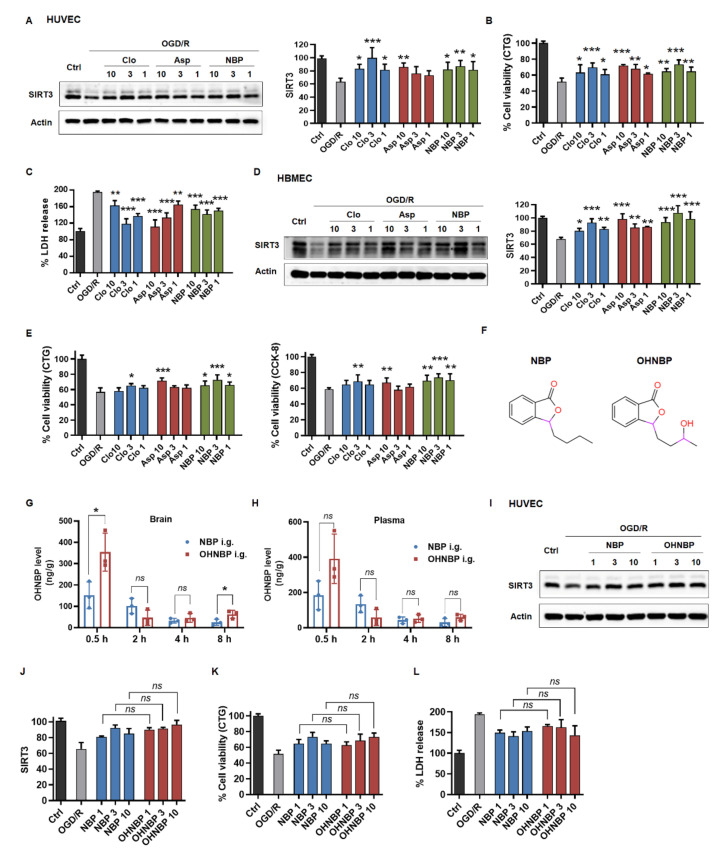
Clinical drugs rescue SIRT3 to reduce OGD/R-induced injury. (**A**) The effect of Clo, Asp and NBP on SIRT3 expression in HUVECs after exposure to OGD/R as assessed by Western blotting. (**B**,**C**) The effect of Clo, Asp and NBP on cell viability in HUVECs after exposure to OGD/R by using CTG assay or LDH kit. (**D**) The effect of Clo, Asp and NBP on SIRT3 expression in HBMECs after exposure to OGD/R as assessed by Western blotting. (**E**) The effect of Clo, Asp and NBP on cell viability in HBMECs after exposure to OGD/R by using CTG or CCK-8 assay. (**F**) Chemical design of OHNBP. (**G**,**H**) The concentration of OHNBP was detected in plasma and brain after 20 mg/kg OH-NBP or NBP was administered to SD rats via oral gavage. (**I**,**J**) The effect of OHNBP and NBP on SIRT3 expression in HUVECs after exposure to OGD/R as assessed by Western blotting. (**K**,**L**) The effect of OHNBP and NBP on cell viability in HUVECs after exposure to OGD/R by using CTG assay or LDH kit. The data are presented as the means ± SD, and were assessed using one-way ANOVA. * *p* < 0.05 vs. the OGD/R group, ** *p* < 0.01 vs. the OGD/R group, *** *p* < 0.001 vs. the OGD/R group. ns, no significance. *n* = 3 for all.

## Data Availability

The datasets in this study are available from the corresponding author upon reasonable request.

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
