# Peer review of "Rescue of Mitochondrial SIRT3 Ameliorates Ischemia-like Injury in Human Endothelial Cells"

_ijms, 2022, doi:10.3390/ijms23169118_

Round 1
Reviewer 1 Report
In the article “Rescue of mitochondrial SIRT3 ameliorates ischemia-like injury in human endothelial cells”, authors aimed to investigate the role of sirtuins in human endothelial cells after induction of ischemic injury.
The article is well structured, the materials and methods are described precisely and in detail, the results are reported concisely and directly.
However, there are some major points to be clarified.
- Although the title refers to endothelial cells in a generic way, in the abstract and in the introduction, there are references to endothelial cells of the cerebral vasculature. The two cell lines used in the study are HUVEC (Human Umbilical Vein Endothelial Cells) and HBMEC (Human Brain Microvascular Endothelial Cells). Is the aim of the research to investigate the role of SIRT3 in endothelial cells in general or in brain endothelial cells specifically? And in the second case, are HUVEC used as a model for the study of brain endothelial cells? If so, the supporting references should be cited.
- Describe the correlation between ischaemic stroke and old age in the introduction, so that the topic is immediately framed. This will also make the understanding of the role of these proteins clearer
- Is there an explanation why cleaved caspase and procaspase 3 were investigated only in HUVEC and not in HBMEC? Why were the SIRT3 overexpression experiments not conducted in both cell lines? These choices need explanation.
- Another unclear point: the published sequence data of profiling brain cortex from rat stroke model showed that SIRT1 had no significant change; the result was confirmed also at mRNA level. Why has overexpression not only of SIRT3 but also of SIRT1 been induced? And why was SIRT1 chosen over SIRT2 and SIRT7, which had also been previously investigated?
- If the mitochondrion is damaged by the initial production of ROS, it is possible that SIRT3 is not able to remedy this problem as it is located within the mitochondrion itself. You could investigate this further to see to what extent the protein can protect this cellular organelle.
- The hypothesis that SIRT3 is possible to exert ischemia-protecting effects through its direct modulation of mitochondrial function present in the discussions is certainly interesting but needs further studies to be confirmed.
- For a more straightforward treatment, it is advisable to make two separate images per cell type to emphasise the differences in the treatments without burdening the same image with a further comparison between different cell lines.
In addition to these major points, some minor points:
- A list with the acronyms used and their explanation is recommended for an easier understanding of the work
- A hint on the correlation between ischemic stroke and old age in the introduction is recommended, so that the topic is immediately framed. In this way the proteins studied will also be better represented as age-regulators.
Reviewer 2 Report
1. page 6 lines 172-181 the correlations of SIRT3 and beneficial effects are not clearly presented.
2. page 8, lines 239-243, these are not clearly presented in the results.
Round 2
Reviewer 1 Report
In the revised article, the authors considered the comments and responded appropriately to all the points, clarifying the purpose of the research and the rationale for conducting the experiments, complementing the results with further experiments and highlighting the need for further studies in the future. With no further revisions to ask, the article is recommended for publication.